# Randomized Controlled Trial Protocol on the Effects of a Sensory Motor Intervention Associated with Kangaroo Skin-to-Skin Contact in Preterm Newborns

**DOI:** 10.3390/ijerph21050538

**Published:** 2024-03-18

**Authors:** Mariane de Oliveira Nunes Reco, Daniele Almeida Soares-Marangoni

**Affiliations:** 1Graduate Program in Health and Development, Faculty of Medicine, Federal University of Mato Grosso do Sul, Campo Grande 79070-900, Brazil; mariane.reco@ebserh.gov.br; 2Graduate Program in Movement Sciences, Institute of Health, Federal University of Mato Grosso do Sul, Campo Grande 79070-900, Brazil

**Keywords:** premature birth, early intervention, sensorimotor interventions, kangaroo mother care

## Abstract

There is still very limited evidence on the effects of neonatal interventions on infant neurodevelopmental outcomes, including general movements (GMs). This research will primarily assess the effects of a sensory motor physical therapy intervention combined with kangaroo skin-to-skin contact on the GMs of hospitalized preterm newborns. Secondary outcomes include body weight, posture and muscle tone, behavioral state, length of hospital stay, and breastfeeding. This study protocol details a two-arm parallel clinical trial methodology, involving participants with a postmenstrual age of 34–35 weeks admitted to a Neonatal Intermediate Care Unit (NInCU) with poor repertoire GMs. Thirty-four participants will be randomly assigned to either the experimental group, receiving a 10-day sensory motor physical therapy associated with kangaroo skin-to-skin contact, or the control group, which will only receive kangaroo skin-to-skin contact. The study will measure GMs (primary outcome), and body weight, posture and muscle tone, behavioral state, length of hospital stay, and breastfeeding (secondary outcomes). Data collection occurs in the NInCU before and after the intervention, with follow-up measurements post discharge at 2–4 weeks and 12–15 weeks post-term. SPSS will be used for data analyses. The results will provide novel information on how sensory motor experiences may affect early neurodevelopment and clinical variables in preterm newborns.

## 1. Introduction

The complications of preterm birth (<37 weeks of gestation) remain the major cause of neonatal mortality worldwide. Even when they survive, millions of preterm newborns still lead their entire lives with developmental problems [1].

Research supports the idea of providing neonatal interventions to improve outcomes in preterm newborns through the benefits of the plasticity of the young nervous system [2]. For example, therapist-delivered postural control interventions in the neonatal intensive care unit (NICU) are effective in promoting gains in the motor behaviors of preterm newborns at hospital discharge [3]. Nevertheless, there is still limited evidence on the effects of interventions in the NICU setting on infant developmental outcomes [3,4]. Investigating the effects of therapist-led interventions delivered to preterm newborns in the NICU is important, as the newborns can spend a long stay in this environment while in a period of maximal neuroplasticity.

A variety of measures have been used to assess early motor outcomes [2]. We wonder whether a sensory motor intervention could influence the newborn’s general movements (GMs) assessed using Prechtl’s General Movement Assessment (GMA). The GMA [5,6] is one of the gold standard tools used to identify early problems in the developing brain [7,8,9,10]. GMs are spontaneous movements performed from the first weeks of fetal life to around 20 weeks post-term. They have been considered excellent markers of neurological deficits as they express brain function [7,8,9,10,11,12,13].

Few studies have examined the effects of early intervention on GMs. Most of them lacked a control group. For example, case studies and pretest–post-test trials have reported improved GM outcomes at 3 months post-term after movement imitation therapy was performed during cramped-synchronized movement—which can signal brain function impairment [14] and increase the temporal organization of GMs after motor interventions [15]. On the other hand, in randomized controlled trials, GMs were not affected by the motor interventions performed before term [16,17,18]. In a 15 min multimodal intervention delivered by a nurse to newborns in the NICU for a few weeks, no effects on the GMs at 3 weeks post-term were found; however, after at least 17 weeks of intervention delivered by the parents at home, the GM rate was higher in treated infants than that in controls [19]. Considering the contrasting findings and the scarcity of randomized controlled trials conducted to guide evidence-based practice on this topic, further trials are needed to investigate the short- and long-term effects of early intervention approaches on the GMs of at-risk newborns. This study will shed more light on the potential role of sensory motor stimuli in changing the quality of GMs and offer better evidence for decision making in the neonatal care unit.

### Rationale for the Intervention

Preterm newborns are unable to maintain postural organization due to their maturation-related hypotonia [20]. This favors the development of extensor muscle retractions, which may lead to motor delay [21] and a reduced quality of spontaneous movements [12,21]. Likely, favoring the maturational adjustment of a newborn’s flexor chain through handling could stimulate their body organization, thus favoring spontaneous global movements [22].

Regardless of the potential influence of musculoskeletal features on spontaneous movements [12,21], GMs express brain function [11]. Based on the recommendations that interventions need to promote active infant movement [2] and improve infant mobility [20], allowing newborns to perform their GMs in a period of reduced mobility during hospitalization might play a role in their neuromotor outcomes. Additionally, multimodal early intervention may change developing neural pathways [23], which perhaps could affect GMs.

Sensory motor stimulation also deserves attention in the NICU. It has been defined as an early intervention comprising a set of strategies designed to enhance neuropsychomotor development by encouraging sensory stimuli tailored to the individual’s functional development, gestational age at birth, and weight. The Brazilian recommendation on physical therapy with sensory motor stimulation in newborns in the NICU has shown that multisensory (auditory, tactile, visual, vestibular) or tactile–kinesthetic stimuli stand out for enhancing behavioral organization, weight and oral behavior, and muscle tone strength or maturation, while skin-to-skin contact enhances behavioral organization, vital physiological parameters, weight and oral behavior, and reduces the length of hospitalization. In general, it is recommended that sensory motor intervention be adapted to the infant’s specific needs and carried out by expert professionals [24].

Tactile experiences, such as those provided with touching [2,25] and with skin-to-skin contact, are particularly important because they have the potential to improve newborn experiences [25]. Skin-to-skin contact is one of the four components of kangaroo care and has become a worldwide standard of care due to its positive effects on infant global health, including the stabilization of neonatal physiological parameters, better state of sleep, weight gain, reduction in the length of hospital stay, breastfeeding and bonding optimization, and increase in cortical activity [26,27,28,29,30]. A guideline developed based on the World Health Organization’s guideline development process has recommended that, irrespective of the mode of birth, immediate, continuous, and uninterrupted skin-to-skin contact should be the standard of care for all mothers and newborns [31].

Although there is evidence of positive effects of neonatal interventions focusing on enhancing movement quality and sensory stimulation to improve infant development [2,3,4,24,25], additional work is needed as physical therapists routinely administer varied techniques to preterm newborns considering the above-mentioned principles.

We aim to investigate the short- and medium-term effects of a sensory motor physical therapy intervention associated with kangaroo skin-to-skin contact, compared to kangaroo skin-to-skin contact solely, on the GMs of preterm newborns in a neonatal unit. The secondary objectives are to determine the effects of the intervention on body weight gain, posture and muscle tone, length of hospital stay, and the establishment and maintenance of breastfeeding. For the technique to be considered beneficial, treated newborns must have a better quality of GMs, or, secondarily, gain body weight, present higher scores for neonatal posture and muscle tone, spend fewer days at the hospital, and present higher proportions of breastfeeding compared to controls.

## 2. Materials and Methods

### 2.1. Study Design and Trial Registration

This is a two-arm parallel group trial. Figure 1 and Figure 2 show the details of the study design.

The trial is registered with the Brazilian Clinical Trials Registry (https://ensaiosclinicos.gov.br/rg/RBR-4wx7wp, accessed on 13 March 2024; date of registration: 27 September 2019; last version: 18 October 2023). First, participants were enrolled on 1 February 2020. There was a break time in the recruitment from March 2020 to June 2021 due to the COVID-19 pandemic restrictions. As of 18 October 2023, we have enrolled 28 newborns. Participant recruitment is expected to be completed by 30 April 2024.

### 2.2. Population

Newborns are recruited from the Neonatal Intermediate Care Units (NInCUs) of the Maria Aparecida Pedrossian University Hospital and the Mato Grosso do Sul Regional Hospital, in the city of Campo Grande, Mato Grosso do Sul (Brazil). To estimate the minimum number of participants, considering a difference of at least 50% between groups for the quality of GMs after intervention using the chi-square test, the suggested sample number is 13 participants per group (80% power; α = 5%). Considering possible losses to follow-up, we will include two groups of 17 participants. The recruitment of an adequate number of participants is ensured by screening every newborn admitted to the NInCU in the hospitals.

The inclusion criteria for starting the interventions are a postmenstrual age of 34 weeks, more than 72 h of postnatal life, admitted to a Neonatal Intermediate Care Unit, with a stable clinical condition, with or without central/peripheral venous access, without any need for invasive or non-invasive mechanical ventilation, and a poor repertoire of GMs (see Section 2.6). Newborns could not present (a) congenital malformations, (b) genetic syndromes, (c) progressive conditions, (d) orthopedic problems, (e) grade III and/or IV peri-intraventricular hemorrhage, (f) hyperbilirubinemia, (g) congenital infections, (h) infections under treatment (change in blood count and positive blood culture), (i) an Apgar score lower than 7 in the fifth minute. All newborns should spontaneously breathe room air, although additional oxygen is not an exclusion criterion. A written, informed legal consent form is to be previously signed by the participant’s parents.

### 2.3. Randomization

Randomization is performed at the beginning of the project by computing two balanced groups using a Matlab routine with a random binary generator (experimental or control). Allocation will be concealed from researchers and participants before the beginning of the intervention in sequentially numbered, sealed, opaque envelopes. The sequence in which the envelopes are opened defines the newborn’s allocation. Randomization, allocation, and concealment will be conducted by a researcher unaware of the study objectives.

### 2.4. Blinding

The primary outcome will be video recorded and assessed posteriorly by two independent assessors, who are blind to the newborn allocation to groups. The medical team who describes secondary outcomes in the medical records will also be blind to newborn allocation as they are not involved in the study. Due to the nature of the intervention, it is not possible to blind the researcher who delivers it to the newborns. The nature of the intervention also prevents the blinding of the posture/muscle tone assessor as they can observe whether the newborn has received physical therapy in the NInCU. Newborns are identified only with their randomization number for data analysis purposes, and data analysts will be blinded to the allocation as far as possible.

### 2.5. Intervention

#### 2.5.1. Physical Therapy Associated with Skin-to-Skin Contact

The physical therapy intervention will be performed between feeding times, always at the same time of day. The experimental group will undergo sensory motor physical therapy associated with the kangaroo skin-to-skin contact in 10 sessions (once a day) throughout 15 consecutive days (intervals on Sundays and when the mother/father is not present). The intervention begins when the newborn reaches 34–35 weeks in postmenstrual age. Therefore, by the end of the interventions (15 days), each newborn has received 10 sessions of intervention (lasting 15 min each) associated with the kangaroo skin-to-skin contact (lasting 60 min) and has reached 36–37 weeks in postmenstrual age.

The physical therapy intervention replicates techniques that have been already used in real NInCU settings [32], including involved hospitals, and consists of sensory motor handling and stimuli (Table 1).

#### 2.5.2. Control Intervention

The newborns in the control group undergo the same procedures as the skin-to-skin contact performed in the experimental group; however, they do not receive the sensory motor physiotherapeutic intervention. Thus, the newborn is maintained in the incubator/crib before undergoing the kangaroo contact with the parent.

The physical therapy intervention and control procedures are performed by a single researcher, who is a physical therapist with experience in performing the described techniques. Stress factors such as noise and light are minimized through the awareness of the NInCU staff during the experimental and control interventions. When signs of sensory overload are detected, such as color fluctuations (pallor and perioral cyanosis, among others), cardiorespiratory changes (bradycardia, irregular breathing, apnea, irregular respiratory rate), changes in state (crying, irritability, startles, hiccups, yawns, salivation), and signs of withdrawal, the intervention is immediately interrupted, and physiological positioning is performed and a firm touch is used to allow the newborn to organize him/herself. The intervention is considered completed if at least 85% of its total is performed. All newborns receive conventional hospital care.

### 2.6. Outcome Measures

#### 2.6.1. Primary Outcome

The primary outcome comprises the differences between groups after the intervention (experimental versus control) regarding the GM outcomes assessed using Prechtl’s GMA.

GMs involve the whole body in a variable sequence of the upper and lower limbs, neck, and trunk and are classified according to age group as (a) fetal and preterm movements, from the fetal period up to 37 weeks of postmenstrual age; (b) writhing movements (WMs), present from 38–40 weeks of postmenstrual age to the 9th post-term week; and (c) fidgety movements (FMs), present from the 9th to the 20th post-term week. Normal WMs are characterized as small to moderate amplitude movements, with low to moderate velocity, that occur elliptically and give the impression of contortions. Their abnormal patterns are classified as poor repertoire (PR)—motor patterns with a monotonous sequence that lack variability; cramped-synchronized (CS)—limb and trunk muscles contract and relax simultaneously, without the fluency or complexity character of normal patterns; and chaotic (Ch)—movements of great amplitude that lack the fluency and elegance of regular motor patterns. Normal FMs are present and characterized by low-amplitude limb, trunk, and head movements of moderate velocity, variable acceleration, and small rotational movements of the hands and feet. Abnormal patterns in this period are classified as absent FMs—no fidgety movements; and abnormal movements—moderate or intense increases in amplitude and velocity and the loss of continuity of FMs [5,6]. Hence, the categories of the quality of GMs comprise Normal, PR, CS, Ch, absent (sporadic) FMs, and abnormal FMs.

During hospitalization, GMs will be recorded using a cell phone following the standards of Prechtl’s GMA [5,6], with the newborns in supine position, using a diaper only or body suit, without pacifiers, and without physical or verbal interaction. Video recordings will last 3–5 min. and are always carried out by the same researcher at 2 time points: (I) on the day before day 1/the beginning of the protocol (34–35 weeks of postmenstrual age); and (II) one day after the end of the protocol (day 16, at around 36–37 weeks of postmenstrual age).

After hospital discharge, GMs will be recorded by the parents and sent to the first author using a cell phone application at 2 time points: (I) at 2–4 weeks post-term (WMs age) and (II) at 12–15 weeks post-term (FMs age). Parents will have been previously instructed by the researcher on how to video record the infants for the GMA at home. Assessments should occur between feeding intervals (around 60–90 min) and not coincide with vaccination days.

The recorded GMs will be assessed by two independent researchers who are certified by the GM Trust. In cases of disagreement, a third certified assessor will be consulted for a final consensus. The assessors will be blind to the newborns’ allocation to the groups.

#### 2.6.2. Secondary Outcomes

The secondary outcomes comprise differences between groups after the intervention regarding body weight, posture behavior and muscle tone, behavioral state, length of hospital stay, breastfeeding success, and maintenance of breastfeeding. Details and time points are provided in Table 2.

### 2.7. Monitoring

As the study is considered low-risk, we do not have a data monitoring committee or specific procedures for stopping the study or interim analyses. Every infant in the NInCU is continuously monitored by the hospital staff as part of routine clinical care. Information regarding vital signs (respiratory rate, heart rate, body temperature, and oxygen saturation), as well as respiratory distress measured using the Silverman–Anderson Bulletin [40], are recorded by the researcher every day from day 1 to day 10 immediately before, immediately after, and 30 min after the intervention (experimental and control). Spontaneous adverse events will be reported to the NInCU staff and the involved Research Ethics Committee and trial registry platform. Signs of clinical deterioration seen during any visit will be reported to the NInCU staff, who will decide on whether the newborns can continue receiving the research interventions.

### 2.8. Data Analysis

Analyses will be performed using SPSS 23.0 (IBM Corp., New York, NY, USA). All data will be entered twice and audited by two researchers before being analyzed. Statistical analysis will be performed according to intention-to-treat principles. Therefore, all included newborns will be analyzed. Data normality will be tested through the visual inspection of histograms and validated using the Shapiro–Wilk test.

Background characteristics in both groups will be presented in a baseline table and summarized as means and standard deviations (continuous variables) or as frequencies and percentages (categorical variables). These data will represent gestational age, age at the beginning of the intervention, birthweight, weight on day 0 of intervention, weight on day 11, the Apgar score, and maternal age.

To test differences between groups (treatment effects) for categorical variables (the quality of GMs, behavioral state, breastfeeding success and maintenance) the chi-square test will be applied. To test differences between groups for body weight, an independent t-test will be used. For other continuous variables, *t*-tests or Mann–Whitney’s test will be used to calculate differences between groups. Adjusted means and 95% confidence intervals can also be calculated using the mixed linear model (GLM) for the continuous variables, considering the interaction terms group x assessment days. Mixed linear models automatically adjust for differences between groups considering differences in the baseline data, even if these differences are very small [41].

A *p*-value < 0.05 will be considered for all analyses. The results will be reported considering the Consolidated Standards of Reporting Trials (CONSORT) statement.

### 2.9. Ethics and Dissemination

The study will be conducted in accordance with the ethical principles of the Declaration of Helsinki and the current Guidelines and Regulating Norms of Research Involving Human Beings of the National Health Council Resolution. Ethics approval for all aspects of this study has been granted (CAAE: 01455818.2.0000.0021, Plataforma Brasil). Written, informed legal consent has been obtained prospectively from the parent/guardian for the newborns’ participation in the study. Protocol modifications will be reported to the involved Ethics Committee and the trial registry platform.

The outcomes of this study will be submitted for publication in a peer-reviewed journal and disseminated via clinical meetings, scientific conferences, and social media.

## 3. Conclusions

Evidence of the effects of neonatal interventions on GMs is very limited. The results of this study will enable us to obtain novel information on how sensory motor experiences may affect early motor behaviors and clinical features in preterm newborns. This is important for expanding our knowledge for decision making on neonatal interventions.

## Figures and Tables

**Figure 1 ijerph-21-00538-f001:**
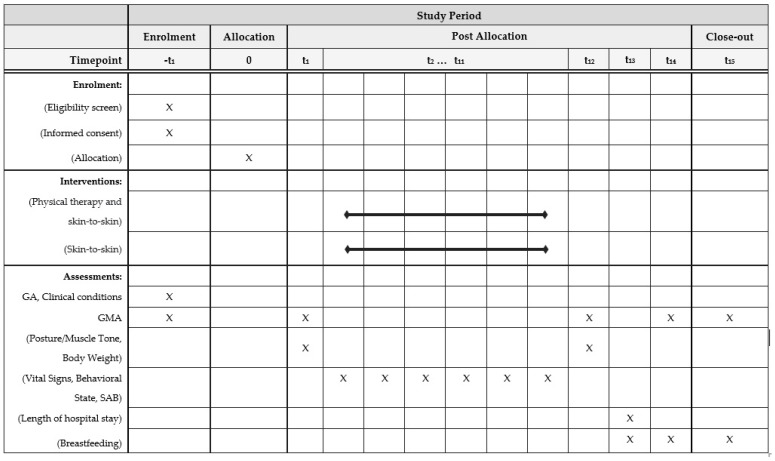
Schematic depicting enrolment, interventions, and assessments (SPIRIT template). GA: gestational age; GMA: General Movement Assessment; SAB, Silverman–Andersen Bulletin; -t_1_: screening period; t_1_: day before the intervention (day 0); t_2_–t_11_: intervention (days 1 to 10); t_12_: day after the intervention; t_13_: hospital discharge; t_14_: 2–4 weeks post-term; t_15_: 12–15 weeks post-term. SPIRIT: Standard Protocol Items: Recommendations for Interventional Trials.

**Figure 2 ijerph-21-00538-f002:**
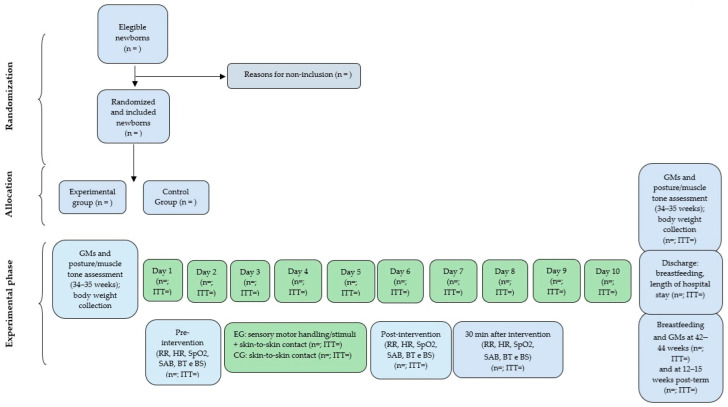
Flow diagram that will be used in the study. GMs: general movements, EG: experimental group, CG: control group, RR: respiratory rate, HC: heart rate, SpO2: oxygen saturation, SAB: Silverman–Andersen Bulletin, BS: behavioral state, BT: body temperature, ITT: intention-to-treat analysis.

**Table 1 ijerph-21-00538-t001:** Descriptions of the sensory motor intervention associated with kangaroo skin-to-skin contact.

Handling and Stimuli	Illustration
(I) Passive mobilization and stretching, which comprise lumbosacral pomp, posterior stretching, cervical stretching, and thoracohumeral dissociation: The physical therapy technique starts with a lumbosacral pomp with the newborn in supine with the head slightly flexed; the physical therapist places their right hand on the newborn’s lumbosacral region while their left hand supports the newborn’ anterior region of the pelvis, gently performing traction in the caudal direction with their right ring and medium fingers [33]. Without losing pelvic positioning in retroversion, the newborn is then moved to lateral decubitus, and posterior stretching is performed, keeping the right hand in the newborn’s sacral region and the left hand positioned under the newborn’s occipital squama, while a subtle and passive traction, always on expiration, is performed by the physical therapist on both sides of the newborn’s vertebral axis [34]. Right after that, the newborn’s cervical region is stretched by the physical therapist’s right hand, involving the newborn’s shoulder, while their left hand supports the newborn’s occipital and temporal region; then, the newborn’s head and neck are slid to the right side; while the left shoulder is gently lowered, the other shoulder is released, and the head is returned to the midline; the hands are then reversed and the movements are performed on the other side [35]. Immediately after the cervical stretching, a thoracohumeral dissociation is performed with the newborn in supine; the physical therapist holds the newborn’s shoulder region with one hand and performs circular movements in a posteroanterior direction [34]. Duration: each handling lasts 2 min (total 8 min), with pauses when necessary, according to the newborn’s tolerance.	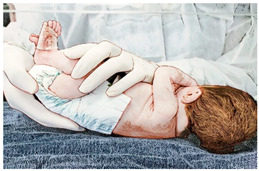 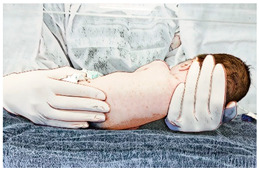 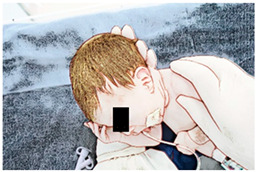 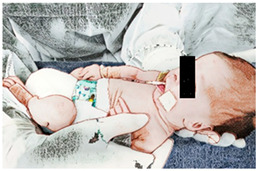
(II) Sensory experience: The newborn remains in supine, the cervical spine remains straight, and the trunk and lower limbs are flexed; the newborn’s hands are passively taken to their parietal region, sliding the palms of the hands anteriorly and laterally on their face, gently moving them on their cheeks, mouth, nose, and eyes [35]. Duration: 2 min.	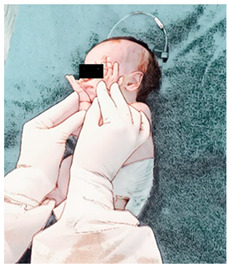
(III) Spontaneous movements: The newborn is positioned in supine in the incubator/crib to be allowed to perform spontaneous movements. Duration: 5 min.	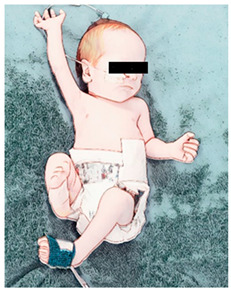
(IV) Kangaroo skin-to-skin contact: Soon after the spontaneous movements, the newborn is placed in the kangaroo position for skin-to-skin contact time with their mother or father. The newborn is placed in the kangaroo position only in diapers in a vertical or diagonally elevated position, head lateralized, upper and lower limbs in flexion and adduction, between the mother’s breasts or between the father’s nipples. Skin-to-skin contact is ensured by not wearing clothes on the thoracic region. The mother or father has no clothes/bra on the chest and wears a hospital gown with an anterior opening. After positioning the newborn, the mother/father wraps the newborn with the gown and are wrapped with a moldable cotton band for greater security [36]. Duration: 60 min.	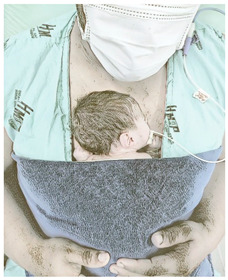

**Table 2 ijerph-21-00538-t002:** Descriptions of secondary outcome measures.

Outcome Measure	Time Point Assessment
Body weight: Number of grams gained or lost between the first and the last day of the intervention, obtained from the medical team’s records on the day before and the day after the end of the intervention. Fifteen grams is expected for preterm infants to gain per day [37].	Body weight is collected at each day of the intervention. The data at the day before day 1 and at the day after the 10th session of the intervention will be considered for the measurement of body weight gain.
Posture and muscle tone: Verified by the neonatal neuromotor screening [33], which is based on Dubowitz et al. [38], which assess posture, passive tonus, active tonus, primitive reflexes, and the postural reactions of righting. The newborn must be in an alert state. The screening starts with the observation of the posture, followed by an assessment of the passive tones of the upper and lower limbs, assessments of the active tone, primitive reflexes, and the postural reactions of righting. The final score is composed of the sum of the scores obtained in each of the items as follows: <15 points (hypotony); 15–19 points (normotonic); and ≥30 points (hypertonia). This assesment is easy to conduct and lasts around 10 min, with the newborn in the crib or incubator [33,38].	The day before day 1 and at the day after the 10th session of the intervention.
Behavioral state: Assessed using the adapted Brazelton’s Neonatal Behavioral Assessment Scale, which classifies behavior by the state in which the newborn is currently in, from 1 to 6. State 1: deep sleep, motionless and regular breathing; State 2: active/light sleep, eyes closed, some body movements; State 3: drowsiness, eyes opening and closing; State 4: quiet alert, minimal body activity; State 5: fully awake, vigorous body movements (active alert); State 6: crying [39].	Each day of the intervention immediately before, immediately after, and 30 min after the intervention (experimental and control).
Length of hospital stay: Number of days of hospitalization, obtained from the medical records.	Hospital discharge.
Breastfeeding success: Considered a dichotomic variable (yes/no) that considers the main key points recommended by the Brazilian Ministry of Health [36] for adequate positioning and attachment during breastfeeding: the newborn’s face facing the breast, with the nose at the level of the nipple; the newborn’s body close to the mother’s, the newborn’s head and trunk aligned; the newborn being well supported; more areola visible above the newborn’s mouth; mouth open enough to capture most of the areola; upper lip turned up, and the lower one turned out; and chin touching the breast. In addition, successful breast milk stimulation must be present. This is assessed by the nursery team and noted in the medical records.	Hospital discharge.
Maintenance of breastfeeding: Defined as the maintenance of exclusive breastfeeding (yes/no). This is assessed by directly asking the mothers	At 2–4 weeks and at 12–15 weeks post-term.

## Data Availability

The raw data supporting the conclusions of this article will be made available by the authors upon request.

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
