# Peer review of "Randomized Controlled Trial Protocol on the Effects of a Sensory Motor Intervention Associated with Kangaroo Skin-to-Skin Contact in Preterm Newborns"

_ijerph, 2024, doi:10.3390/ijerph21050538_

Round 1

Reviewer 1 Report

Comments and Suggestions for Authors

Review

Dear authors

Congratulation for your presented study design.The topic is of great interest. The aim of this study protocol is to emphasize the way you will try to asses the relationship between skin-to-skin kangoroo and sensory motor physical therapy and the neurodevelopment in preterm newborns.

I think the introduction should shed more light on the necessity of this study. You showed that few studies have examined the effect of early intervention on GMs. What kind of studies?[line48]

Material and method are correctly presented and the expected results are of real interest.

Author Response

Thank you very much for taking the time to review this manuscript. Please find the detailed responses below and the corresponding revisions/corrections highlighted/in blue in the re-submitted files.

Review 1: I think the introduction should shed more light on the necessity of this study. You showed that few studies have examined the effect of early intervention on GMs. What kind of studies?[line48]

Response: Thank you for your suggestion. We agree with you. Please find further justification for the study in the lines 49 to 65 of the manuscript.

Reviewer 2 Report

Comments and Suggestions for Authors

WHO has recommended minimum duration of KMC for 8hrs. KMC duration should be increased in your study. Total cumulative hrs of sensory motor stimulation is also less to draw a major impact. Primary outcome measures at 2 to 4 weeks and 12 to 15 weeks postterm physical assessment should be done instead of video recording by the parents

Author Response

Thank you very much for taking the time to review this manuscript. Please find the responses below:

Review 2: WHO has recommended minimum duration of KMC for 8hrs. KMC duration should be increased in your study. Total cumulative hrs of sensory motor stimulation is also less to draw a major impact. Primary outcome measures at 2 to 4 weeks and 12 to 15 weeks postterm physical assessment should be done instead of video recording by the parents.

Response: Thank you for your suggestions. We agree with you that KMC duration should be expanded and will consider that in futures trials. In the current study, it is not possible to change aspects of the intervention at this point because it has already started, as mentioned at the beginning of the Methods section (lines 121-125). It is also important to highlight that the intervention procedures were designed to mimic as much as possible the practice strategies that have already been used in the daily routine of the hospitals. We need to test their efficacy the way it has been delivered to newborns in real scenarios. This was mentioned in lines 229-230 of the manuscript: "The physical therapy intervention replicates techniques that have been already used in real NICU settings [32], including in the involved hospitals...").

Regarding the video recordings by the parents, this strategy will be used when the baby will be already discharged from the hospital, to mitigate data loss. Many parents live in different cities than the hospital and, thereby, in these circumstances the physical assessment would not be possible.